# The packing density of a supramolecular membrane protein cluster is controlled by cytoplasmic interactions

Elisa Merklinger[1], Jan-Gero Schloetel[1], Pascal Weber[1], Helena Batoulis[1], Sarah Holz[1], Nora Karnowski[2], Jérôme Finke[1], Thorsten Lang[1]*

[1]Membrane Biochemistry, Life and Medical Sciences (LIMES) Institute, University of Bonn, Bonn, Germany; [2]Chemical Biology, Life and Medical Sciences (LIMES) Institute, University of Bonn, Bonn, Germany

**Abstract** Molecule clustering is an important mechanism underlying cellular self-organization. In the cell membrane, a variety of fundamentally different mechanisms drive membrane protein clustering into nanometre-sized assemblies. To date, it is unknown whether this clustering process can be dissected into steps differentially regulated by independent mechanisms. Using clustered syntaxin molecules as an example, we study the influence of a cytoplasmic protein domain on the clustering behaviour. Analysing protein mobility, cluster size and accessibility to myc-epitopes we show that forces acting on the transmembrane segment produce loose clusters, while cytoplasmic protein interactions mediate a tightly packed state. We conclude that the data identify a hierarchy in membrane protein clustering likely being a paradigm for many cellular self-organization processes.

*For correspondence: thorsten. lang@uni-bonn.de

**Competing interests:** The authors declare that no competing interests exist.

## Introduction

Molecular components of cells are organized on the large scale through compartmentalization, but also on the nm- to µm-scale via self-organization of molecules into clusters. For the clustering of cytosolic or membrane components, interactions must be strong enough to mediate clustering, but at the same time they should be weak enough to prevent irreversibly forming, cytotoxic aggregates.

A prominent example of such self-organization occurs in the cell membrane, where membrane proteins form microdomains. In the last decade, super-resolution microscopy revealed that such domains are spherical structures with dimensions in the 100 nm range (*Lang and Rizzoli, 2010*), frequently circumscribed as protein clusters. The identified mechanisms underlying clustering are mostly based on forces mediated by the membrane leaflets. Such include hydrophobic mismatch, lipid wetting, protein-lipid ionic sequestering and depletion attraction forces (for review see *Destainville et al., 2016*; *Recouvreux and Lenne, 2016*). However, it was observed that protein isoforms locate to separate clusters (*Uhles et al., 2003*; *Kai et al., 2006*; *Low et al., 2006*), difficult to explain with forces exclusively propagated by the membrane. Therefore, also highly specific protein-protein interactions must contribute to the clustering process.

The question arises whether all these mechanisms additively shift the equilibrium between free and clustered molecules towards the more clustered state, or whether clustering is a step-wise process with specific mechanisms responsible for defined stages. For example, pre-clustering could be mediated via less specific interactions between the small membrane embedded protein segment and its environment. Then, highly specific interactions involving large protein domains could determine the conformational arrangement of the molecules.

At present, for most protein clusters not much more is known than their size and composition. One exception is the cluster formed by the SNARE-protein syntaxin 1A, for which a quantitative model is available (*Sieber et al., 2007*; *van den Bogaart et al, 2013*). Several approaches show that syntaxin nano-clusters are composed of 50–75 molecules, have a size of 50–60 nm and exchange molecules with each other (*Sieber et al., 2007*; *Rickman et al., 2010*; *Knowles et al., 2010*; *Barg et al., 2010*). Syntaxin domains disperse when cholesterol is depleted (*Lang et al., 2001*; *Chamberlain et al., 2001*), are dependent on ionic interactions between PIPs (phosphatidylinositol phosphates) and a polybasic stretch that can disrupt (*Murray and Tamm, 2009*, *2011*) or promote (*van den Bogaart et al, 2011*) clustering, and hydrophobic mismatch is also implicated in clustering (*Milovanovic et al., 2015*).

Hence, forces acting on the transmembrane segment (TMS) play a pivotal role in syntaxin clustering. In fact, the TMS of syntaxin is capable of forming clusters on its own (*Sieber et al., 2006*). However, the SNARE-motif of the cytoplasmic domain targets free syntaxin molecules to already existing syntaxin clusters and thus slows down syntaxin mobility (*Sieber et al., 2007*, *2006*). This suggests that cytoplasmic interactions increase the time a syntaxin molecule resides in a clustered structure, before it is again released and diffuses to the next cluster. It has been speculated whether these interactions have any consequences on the inner architecture of syntaxin clusters as two different molecule arrangements are conceivable: one assuming that molecules are spaced by a few nanometres, and another one in which the SNARE-motifs of the syntaxins are in direct contact (*Sieber et al., 2007*). For differentiating between these scenarios, differences in inter-molecule distances in the nanometre-range must be resolved in a large crowd of molecules. Such resolutions are currently not achievable by super-resolution microscopy as PALM/STORM that allow for single molecule localization with a precision in the 10 nm range. Even if resolutions of 1–2 nm were achieved, it would be difficult to label all the molecules without employing GFP that due to its own physical extension precludes distances shorter than its barrel extensions (2.4–4.2 nm; *Ormö et al., 1996*). Moreover, inter-fluorophore distances in the nm range would cause strong self-quenching, diminishing the signal required for molecule localization.

Here we tackle the question from several angles applying methods that are capable of sensing loose and tight molecular packing in the nm-range. We find that the N-terminal segment of the SNARE-motif is sufficient for tight packing of syntaxin molecules which form loosely packed clusters in the absence of cytoplasmic interactions. A hierarchical model of syntaxin clustering is discussed.

## Results

Syntaxin 1A has a large N-terminal domain, followed by a linker region and a SNARE-motif connected to a TMS (*Figure 1*). Many studies investigating syntaxin clustering attach the GFP label to the molecule's C-terminus. Then, the lateral distribution is investigated directly by super-resolution imaging of fixed samples (*Rickman et al., 2010*), or indirectly in live cells by interpreting tracks of individual molecules (*Barg et al., 2010*) or syntaxin's apparent lateral diffusion (*Sieber et al., 2007*) in terms of clustering behaviour. In neuronal cells, cytoplasmic interactions restrict the mobility of the molecule, which may either indicate a role of the cytoplasmic domain in the clustering process (*Sieber et al., 2007*), or reflect interactions with syntaxin's neuronal binding partner SNAP25 (*Ribrault et al., 2011*). For better clarification, we study the role of the cytoplasmic domain in a non-neuronal cell line (HepG2 cells) that expresses neither syntaxin 1A nor SNAP25. Two deletion constructs were compared to wild-type syntaxin: syx-ΔCyt, lacking essentially the entire cytoplasmic region, and syx-ΔS, from which a N-terminal portion of the SNARE-motif is deleted (*Figure 1*). The latter construct is generated to narrow down the interacting segment to the SNARE-motif, as previous findings proposed its involvement in homophilic interactions for clustering (*Sieber et al., 2007*).

It has been reported that syntaxin expressed in non-neuronal cells does not efficiently reach the cell membrane, unless Munc18 is co-expressed (*Rowe et al., 1999*). To estimate the plasma membrane targeting efficiency of our constructs, we quantified the fraction of the expressed proteins in the plasma membrane by acidic pH-quenching of the fluorescence of their GFP molecules (*Patterson et al., 1997*) that are extracellularly exposed. In addition, we used as reference points GFP located at the cytoplasmic site (GFP-SNAP25) that should be resistant to pH changes of the extracellular environment, and a GFP in the endocytic pathway (VAMP8-GFP; *Antonin et al., 2000*) also located in the lumen of acidic organelles. To reveal GFP fractions in acidic organelles, we

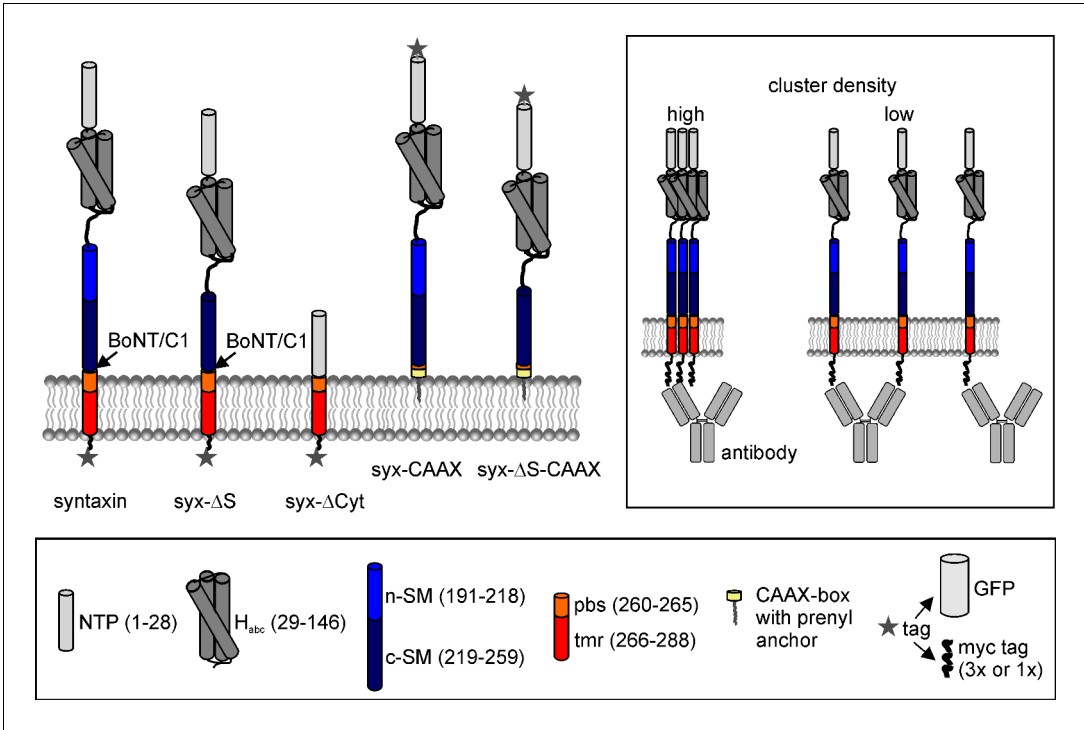

**Figure 1.** Domain structure of syntaxin 1A and deletion constructs. Pictogram illustrating the constructs' domain structure, the botulinum neurotoxin C1 (BoNT/C1) cleavage site and the tag positions. In case of the CAAX constructs, the TM domain and parts of the pbs were exchanged for a CAAX box from K-Ras that after prenylation anchors the construct to the plasma membrane. Lower legend; N-terminal short peptide (NTP, grey), globular N-terminal domain ($H_{abc}$; dark grey), SNARE motif section divided into an N- and a C-terminal part (n-SM and c-SM in blue and dark blue, respectively), polybasic stretch (pbs; orange) and the remaining transmembrane region (tmr; red), and the prenylated CAAX-box (yellow). Two types of tags were used, a monomeric variant of GFP or a triple myc-tag, both fused via a 12 amino acid linker to the extracellular site. The myc-tags were attached to the extracellular site to minimise the possibility of intracellular conformational changes masking the epitope of the myc-tag. The CAAX constructs carry the myc-tag at the N-terminus and were comparatively studied with N-terminally myc-tagged syntaxin (not shown). Upper right box, accessibility of the antibody to the myc-tag depends on the molecule packing density.

applied bafilomycin that specifically inhibits V-ATPases (*Bowman et al., 1988*), thereby dissipating pH gradients between lysosomal/endosomal compartments and the cytosol. As shown in *Figure 2*, we find a slight drop of fluorescence for GFP-SNAP25, possibly due to small cell lesions induced by the DMSO solvent (which is observed at concentrations above 0.5%; *Qi et al., 2008*). VAMP8, as expected, has a large fraction of intracellular fluorescence that increases further after bafilomycin treatment. The non-quenched fluorescence of the syntaxin constructs ranges between 20–40% and does not significantly increase after bafilomycin treatment. When the intracellular distribution was alternatively analysed by super-resolution microscopy on fixed cells, the variability between the constructs was reduced, which may indicate that optical sectioning microscopy is less capable than pH quenching for resolving intracellular and plasmalemmal fractions (*Figure 2—figure supplement 1*). Collectively, the data show that not all overexpressed proteins reach the plasma membrane. However, there are only minor differences in the plasma membrane targeting efficiency between the syntaxin variants.

When studying the mobility of the constructs by fluorescence recovery after photobleaching (FRAP), we find increased mobility in the order syntaxin, syx-ΔS and syx-ΔCyt (*Figure 3*). In these experiments we observed no relation between expression level and mobility for syx-ΔS and syx-ΔCyt, while syntaxin apparently displays a trend towards lower mobility the higher it is expressed (*Figure 3d*). As the analysed syntaxin-GFP cells tended to be dimmer the effect of the intact

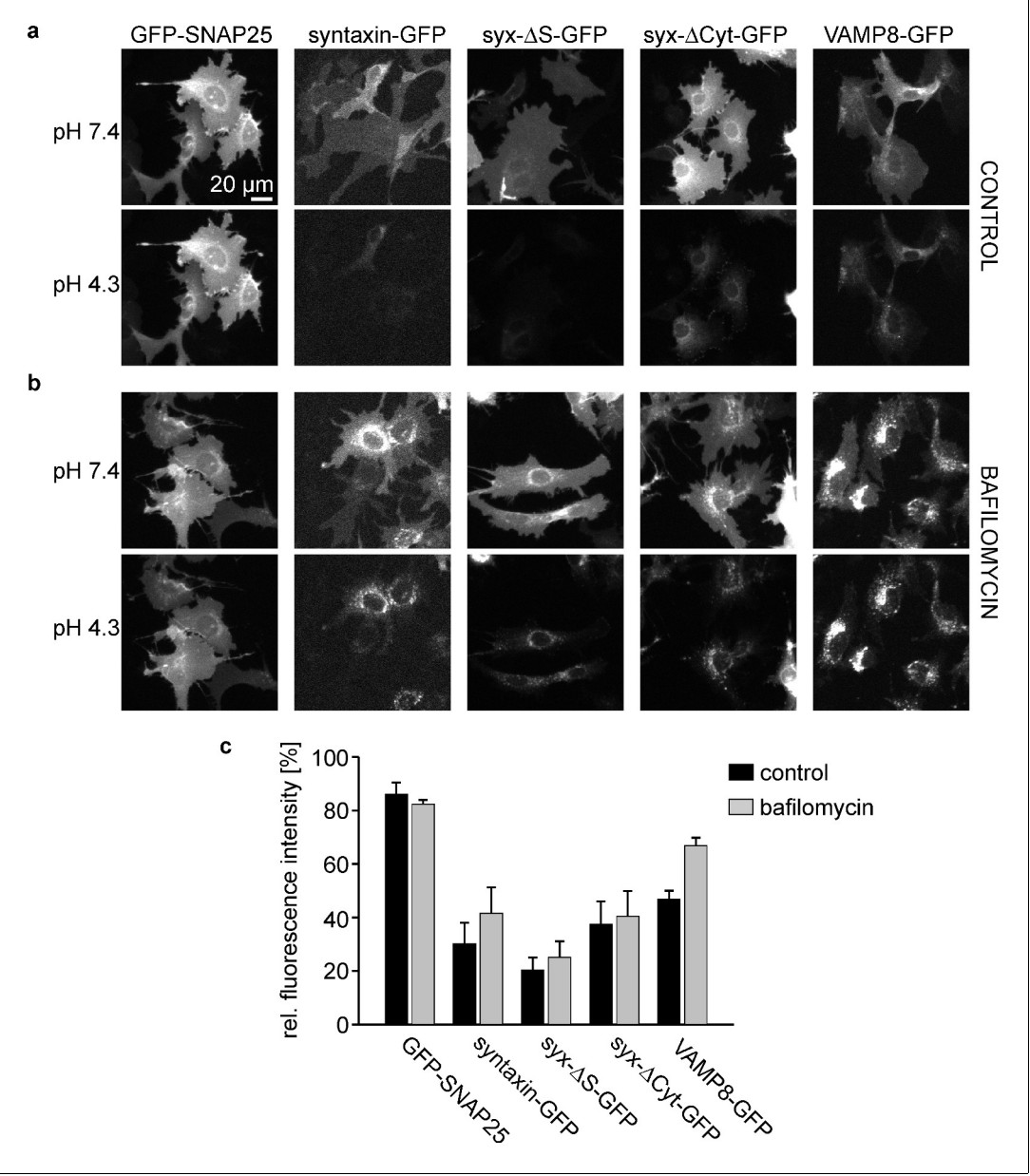

**Figure 2.** Plasma membrane targeting of the different syntaxin variants. HepG2 cells expressing GFP-labelled SNAP25, syntaxin, syx-ΔS, syx-ΔCyt, or VAMP8 were incubated at 37°C for 45 min (**a**) without or (**b**) with 0.2 μM bafilomycin. The GFP-signal was imaged at pH7.4, followed by another image taken at pH4.3. Images in one column are shown at the same scaling. (**c**) Remaining cell fluorescence upon acidification is expressed in percent. Values are given as means ± S.E.M. (n = 3 independent experiments; for one experiment values from 16 to 38 cells per construct and condition were averaged).
The following figure supplement is available for figure 2:

**Figure supplement 1.** Subcellular distribution analysis by STED microscopy.

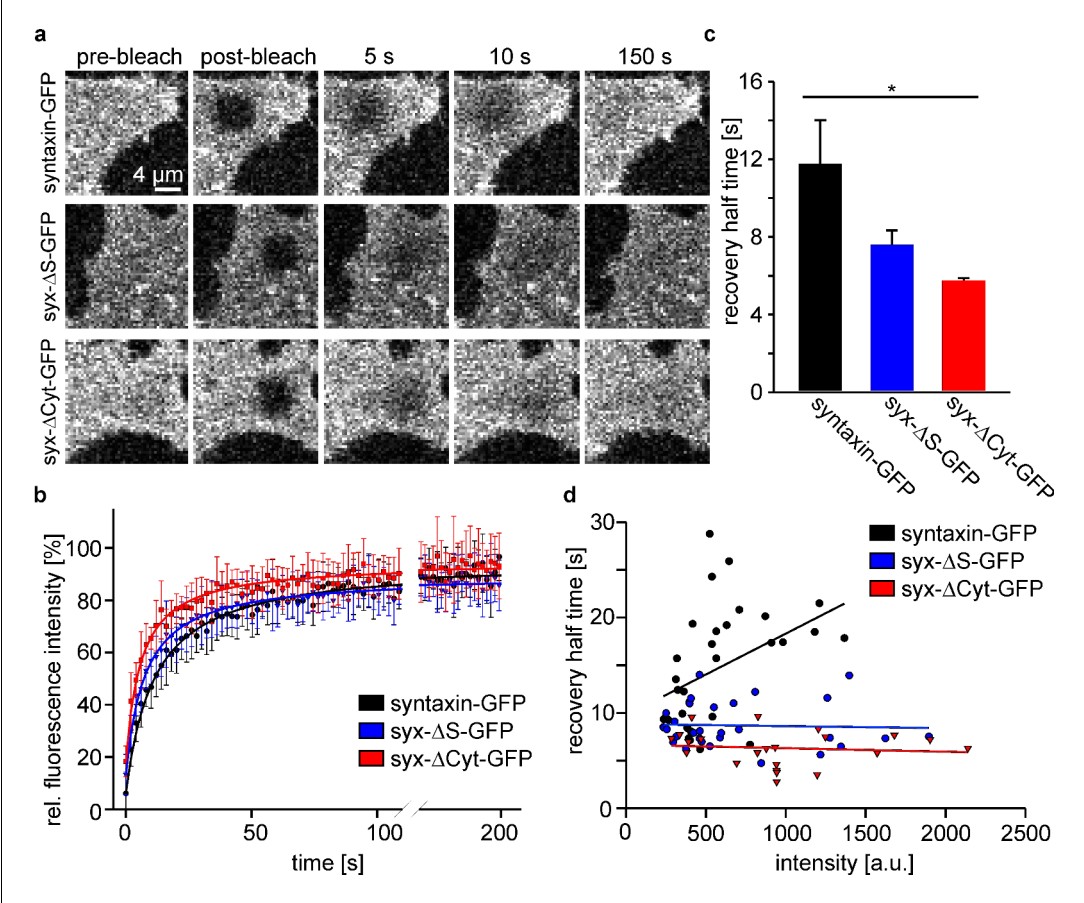

**Figure 3.** The intact syntaxin SNARE-motif is required for mobility restriction in HepG2 cells. Comparative fluorescence recovery after photobleaching analysis of GFP-labelled full length syntaxin and the deletion constructs syx-ΔS and syx-ΔCyt (for details see *Figure 1*). (a) Live cells expressing the indicated constructs are imaged by confocal microscopy at the basal plasma membrane. In a squared region of interest (ROI) fluorescence is bleached (compare pre-bleach and post-bleach images). Then images are taken at 1 Hz for several minutes to monitor the recovery of fluorescence in the ROI whose half-time is inversely proportional to the apparent lateral diffusion coefficient (or molecule mobility). Images are shown at the same scalings. (b) Averaged recovery traces from several cells imaged for one experiment. Values are given as means ± S.D. To better illustrate the traces, only every second point of the measurement is shown. (c) Half-times of recovery obtained by fitting hyperbola functions to traces as shown in (b). Values are given as means ± S.E.M. (n = 3 independent experiments; Kruskal-Wallis one way analysis of variance on ranks (p=0.011; *p<0.05); 8–18 cells per construct and experiment). (d) Half-times of recovery were also determined for individual cells. Values obtained from all experiments were plotted versus the pre-bleach intensity (a relative measure for the expression level). Only for full length syntaxin at higher expression levels a trend is observed towards longer half-times of recovery.

SNARE-motif on mobility restriction (Figure 3c) is potentially underestimated. In addition to live cells, we also studied the constructs in isolated basal plasma membrane sheets. In this preparation, recovery from intracellular structures can be excluded. We made the same two observations. First, constructs become faster in the same order as in cells, and second, only the mobility of syntaxin tends to be slower at higher expression levels (*Figure 4*). Therefore, we can safely conclude that also in non-neuronal cells, independent from interactions with SNAP25, an intact SNARE-motif is required for restricting the mobility of syntaxin.

Though the GFP-molecule was proven to be a suitable tool for studying aspects of syntaxin-clustering, interference with the clustering process cannot be entirely excluded as its physical extensions would impede distances between syntaxin's C-termini shorter than ≈2.5 nm. To exclude such effects we used the smaller and less globular myc-tag (*Figure 1*) supposed to generate none or less steric stress while enabling the use of antibodies as probes for packing density (*Batoulis et al., 2016*). If densely packed, an antibody bound to one myc-tag will prevent binding of additional antibodies to

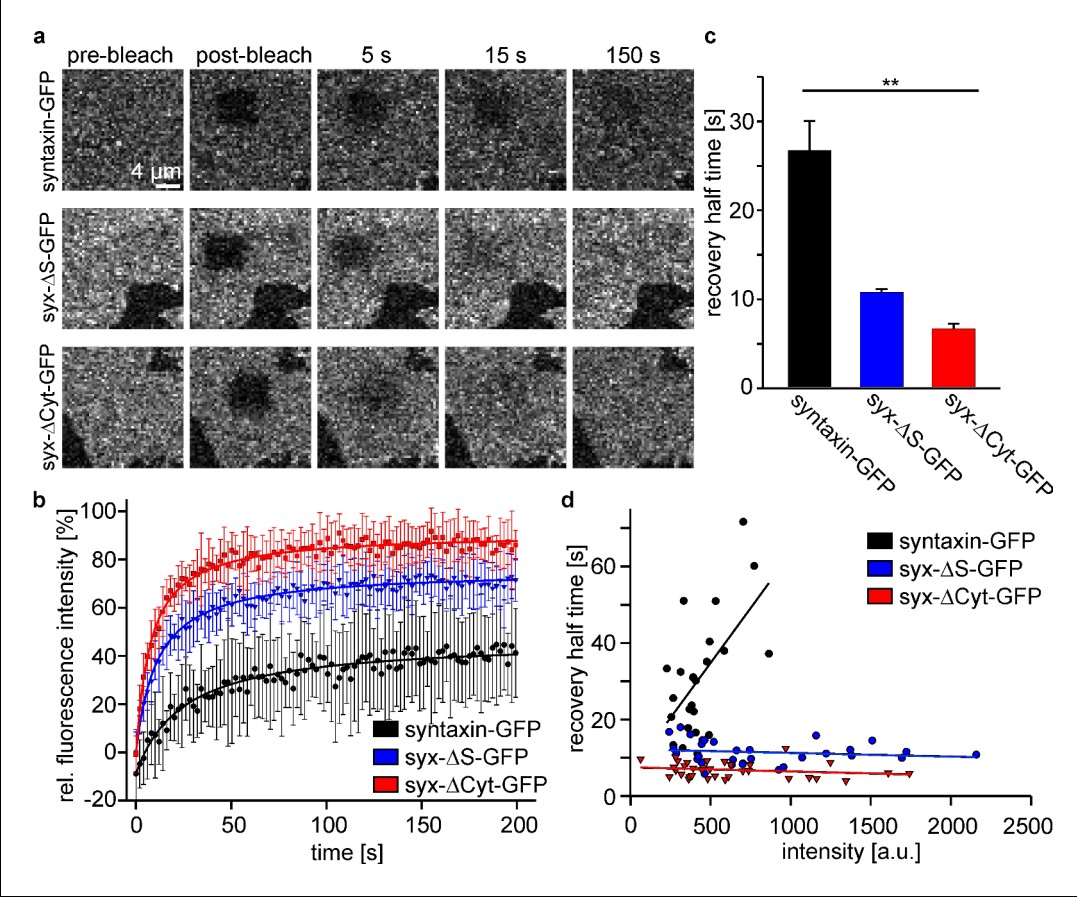

**Figure 4.** FRAP on membrane sheets shows the same mobility behaviour as in intact cells. (**a**) Membrane sheets from cells expressing the indicated constructs were used in FRAP experiments as described in *Figure 3*. Images are shown at the same scalings. (**b**) Averaged recovery traces from 9 to 15 membrane sheets imaged for one experiment. Values are given as means ± S.D. Only every second measurement point is shown (**c**) Half-times of recovery. Values are given as means ± S.E.M. (n = 3 independent experiments; Kruskal-Wallis one way analysis of variance on ranks (p=0.004; **p<0.01); 9–17 membrane sheets per construct and experiment). (**d**) Half-times of recovery from all individual membrane sheets collected from all experiments, plotted versus the pre-bleach intensity. As in intact cells, only full length syntaxin tends to longer half-times of recovery at increased intensity levels.

the neighboured myc-tags (*Figure 1*). Hence, substoichiometric labelling is increased the closer the epitopes are packed in the clustered structures. Likewise, shielding can be attenuated upon looser packing.

First, we investigated the lateral distribution of the myc-tagged constructs by super-resolution microscopy (for subcellular distribution analysis see *Figure 2—figure supplement 1*). Because high resolutions in STED microscopy require strong staining intensities, we applied a classical first and secondary antibody staining. As HepG2 cells lack endogenous syntaxin, starting from a zero background the influence of the expression level on clustering can be studied over a wide dynamic range. We find that cluster number per area strongly depends on the average staining intensity of the membrane and is in the order of magnitude reported previously for neuronal cells (14–19 clusters/μm$^2$; *Sieber et al., 2007*; *Bar-On et al., 2012*), whereas increase in cluster intensity and size is less dependent on the average staining intensity (*Figure 5c*). Spherical clusters are always observed for syntaxin that never yielded as high staining intensities as the other constructs. For syx-*ΔS*, at high intensities we found also elongated structures or even large areas covered by rather homogenous fluorescence (*Figure 5—figure supplement 1*).

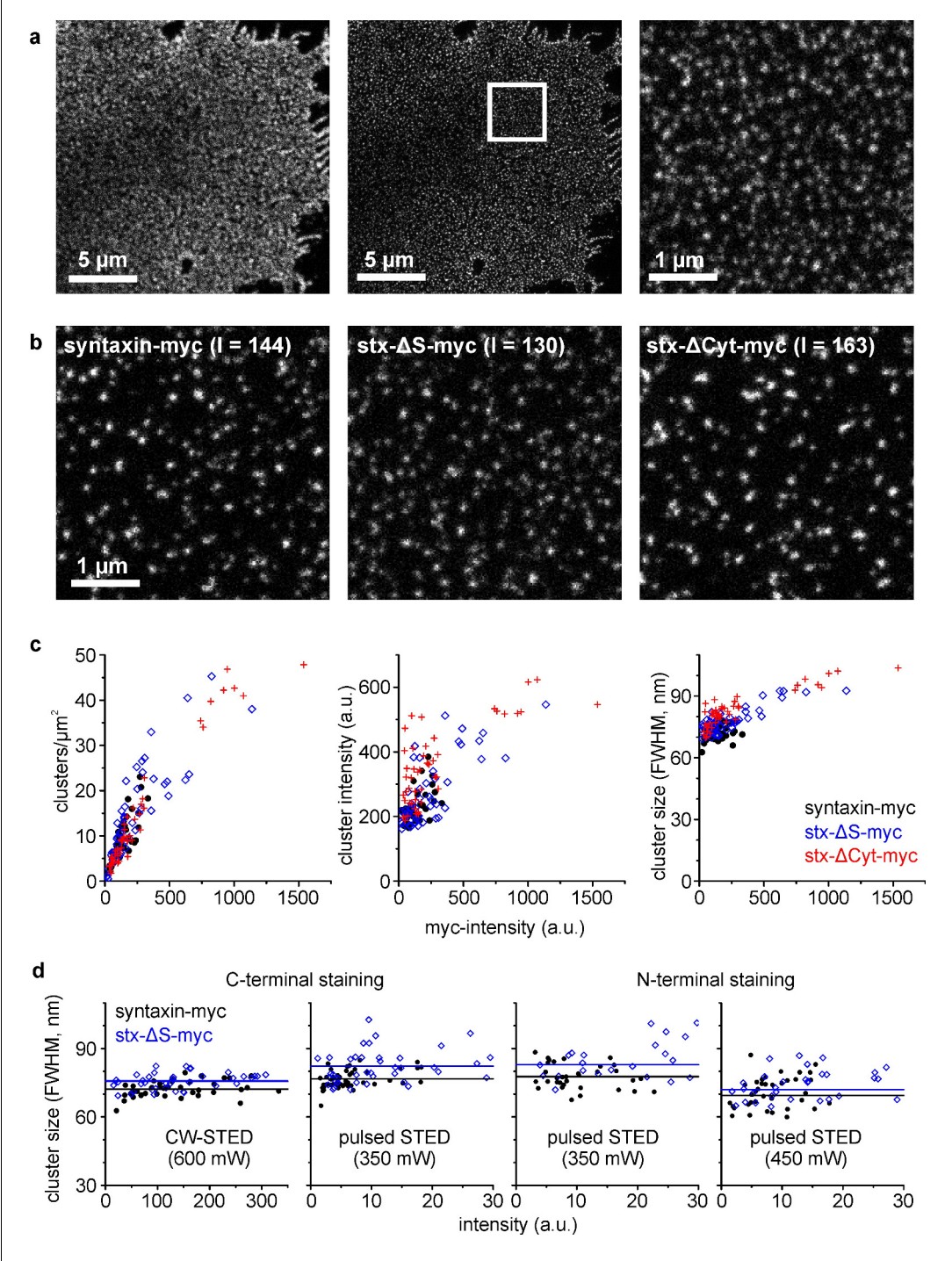

**Figure 5.** Clusters analysed by super-resolution microscopy. (**a**) Plasma membrane with stained syntaxin-myc. Left and middle, images taken in the confocal and STED mode, respectively. A region of interest (white box; for magnified view see right panel) was defined for analysing the image intensity in the confocal image (used for plotting in **c**) and the cluster characteristics in the STED image. (**b**) Images from the different constructs as indicated, chosen with similar intensities (I = intensity; values are given in a.u. in the image). (**c**) Clusters per area, cluster intensity and cluster size, plotted versus the average image intensity. (**d**) Average cluster size for syntaxin-myc and syx-ΔS-myc determined under variable experimental conditions. As indicated, samples were stained with antibodies recognizing either the extracellular C-terminus (anti-myc; left graph is a zoom-in view of the right plot in **c**) or the intracellular N-terminus (anti-HPC1). Apart from CW-STED with a green dye and 600 mW depletion laser intensity (left panel), pulsed STED with a red dye was employed at 350 or 450 mW depletion laser intensity. Only the overlapping intensity range is shown. Black (syntaxin-myc) and blue (syx-ΔS-

*Figure 5 continued on next page*

*Figure 5 continued*

myc) lines indicate the average size. (**c and d**) Each data point represents one membrane sheet (size and intensity are average values of all clustered structures identified on one membrane sheet). Values from three independent experiments are pooled.

The following figure supplement is available for figure 5:

**Figure supplement 1.** Clustering behaviour of syx-*Δ*S.

At first glance constructs behave similarly, with a tendency towards larger and brighter clusters upon shortening or deleting of the SNARE motif (*Figure 5c*). The small trend in size difference between syntaxin and syx-*Δ*S clusters was also noted when a different dye for STED de-excitation is used, and also when, instead of the myc-tag the N-terminal domain was stained (*Figure 5d*). Albeit the absolute measured sizes depend on the label and the intensity of the de-excitation laser, we still can conclude that clusters tend to increase their size upon shortening of the SNARE-motif.

There are two possible explanations for the comparatively dim staining intensities of the syntaxin-myc construct: it may either be expressed at lower levels or the myc epitopes are less accessible for antibodies than in the other constructs. To examine this further, we comparatively studied trans-fected cells by microscopy and western blot: while in microscopy epitope accessibility scales with packing density, for western blot analysis cells are lysed, wherefore it is insensitive to differences in packing density. When relating the immunostaining signal of syntaxin and syx-*Δ*S to their respective western blot signal we find a roughly three-fold brighter staining for syx-*Δ*S (*Figure 6*).

For further evaluation of the epitope accessibility approach, we tested whether staining can be increased after cleaving off the SNARE-motif by the clostridial toxin botulinum neurotoxin C1 (BoNT/C1). Syx-*Δ*S has the same cleavage site, yielding an identical C-terminal cleavage product (*Figure 1*). However, as syx-*Δ*S is less clustered from the beginning, co-expressed BoNT/C1 should affect staining of syntaxin stronger in comparison to syx-*Δ*S. As shown in *Figure 7*, BoNT/C1 does not change staining for syx-*Δ*S-myc, but leads to a significant increase of syntaxin-myc staining. This supports the idea that the cytoplasmic domain, specifically the SNARE-motif, packs molecules denser and thereby decreases the staining intensity of the extracellular epitopes.

A crystal structure analysis of the ternary SNARE-complex showed that the helical-conformation of syntaxin extends beyond the protein's SNARE-motif and continues until its C-terminus (*Stein et al., 2009*). It is thus possible that in live cells the SNARE-motif section, pbs and TMR (*Figure 1*) form one straight α-helix, too, and that such a conformation is a prerequisite for tight packing. To test whether preventing this continuous helix would affect clustering, we replaced the pbs-TMR segment by a CAAX region (syx-CAAX; for structure see *Figure 1*). We compared this syx-CAAX construct to both syntaxin and syx-*Δ*S with a CAAX membrane anchor (syx-*Δ*S-CAAX). All constructs were organized in clusters, and both syx-CAAX and syx-*Δ*S-CAAX were much more accessible for antibody staining than syntaxin (*Figure 8*). We therefore propose that straight extended helices are advantageous for tight protein/cluster packing.

## Discussion

Here, we apply three approaches for probing the packing density of syntaxin clusters. One employs GFP labelling with the advantage that all syntaxin-molecules are visualized. Though the extracellular GFP-tag has no access to the intracellular domains, it still could prevent clustering of molecules at distances shorter than 2.4 nm. In particular, if clustering is driven by SNARE-zippering with the helical conformation of the SNARE-motif extending into the membrane, extracellular GFP would cause mechanical tension. In neuronal cells this should be less of a problem, as endogenous syntaxin molecules mix with GFP-labelled syntaxin in the same cluster, increasing the space between different syntaxin-GFP molecules. Perhaps GFP is the reason why the half-time of recovery is shorter for syntaxin in HepG2 cells (lacking endogenous syntaxin) when compared to PC12 cells (*Sieber et al., 2007*). In any case, syx-*Δ*S molecules move faster than syntaxin molecules.

When myc-tagging is employed, the antibody signal per myc-tag increases by ≈3-fold upon partial deletion of the SNARE-motif. This finding has two implications. First, it indicates that clustered syx-*Δ*S molecules are more spaced. Second, when evaluating the staining pattern of syntaxin and

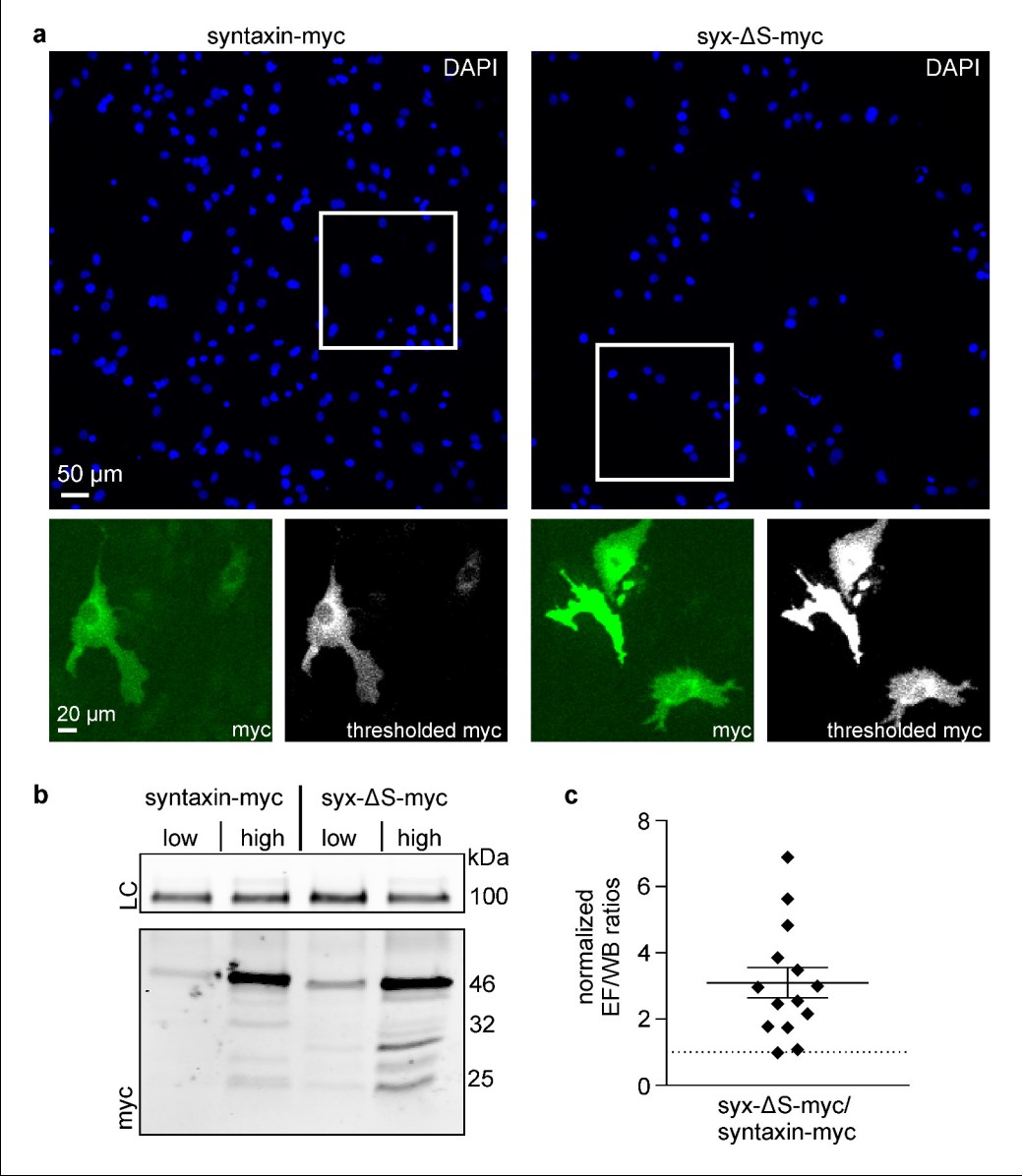

**Figure 6.** Partial SNARE-motif deletion increases antibody access to the C-terminal myc-tag. Cells were transfected with plasmids encoding for syntaxin-myc or syx-ΔS-myc employing low or high plasmid concentrations and each transfection was split into two fractions (one for microscopy and one for western blot analysis). After one day the 'low' and 'high' syntaxin-myc/syx-ΔS-myc pairs were subjected to comparative analysis by microscopy and western blot. (a) For microscopy, fixed cells were stained for their nuclei by DAPI (upper panels represent large overviews of the blue channel) and for the myc-tag by immunostaining (green channel; lower left panels show magnified views from the boxed regions in the blue channel). Using mock transfected cells we set a threshold value to eliminate the offset of background signal. The lower right black-and-white images show the background corrected images. The integral of the thresholded myc-signal was determined and related to total number of cells, yielding the average myc-signal in epifluorescence (EF). (b) For western blot analysis, equal amounts of total protein were loaded for analysis by SDS-PAGE and western blotting, using transferrin receptor as a loading control (LC). Myc-band intensities were quantified and normalized to the respective transferrin receptor loading control, giving WB. (c) Strong variability of the absolute values both in microscopy and western blot analysis yielded highly variable absolute values of the EF/WB ratios (for details see *Figure 6—figure supplement 1*). Therefore, for each of the syntaxin-myc/syx-ΔS-myc pairs, the ratio of syx-ΔS-myc was related to the ratio of syntaxin-myc, yielding the 'normalized EF/WB ratios' plotted in (c). Dotted line indicates a value of 1 (no effect). Values are shown as a dot plot with the mean ± S.E.M. (n = 14 transfection pairs from 28 transfections; statistics was performed on the non-normalized ratios (see *Figure 6—figure supplement 1*).

*Figure 6 continued on next page*

*Figure 6 continued*
The following figure supplement is available for figure 6:

**Figure supplement 1.** Ratios calculated from the absolute signals in microscopy (EF) and western blot (WB) analysis.

syx-ΔS in STED microscopy, image pairs should be compared in which syx-ΔS staining is three-fold brighter. Otherwise, as we currently do, one underestimates the size increase in syx-ΔS clusters.

In conclusion, mobility measurements, cluster-size and myc-epitope accessibility suggest that deletions of the cytoplasmic region produce a state of looser packing. This means that the number of syx-ΔS molecules accommodated in a cluster phase is not limited by space or packing density, which is of conceptual significance. Importantly, in the presence of the full cytoplasmic domain the same copy number of molecules accumulate in fewer clusters, necessarily yielding a higher packing density.

Interestingly, though absolute figures on size are difficult to obtain, no matter which construct and at which expression level cluster size seems to be in a range from 70 to 100 nm, which is also the typical size of other membrane protein clusters (*Lang and Rizzoli, 2010*). Factors restricting the further growth of nano-cluster phases are likely very basic mechanisms, including a combination of cholesterol effects, hydrophobic mismatch and ionic lipid-protein sequestering (*Recouvreux and Lenne, 2016*).

Though at this point we can only speculate on the exact nature of the clustering reaction, two parallels between the SNARE-clustering reaction and SNARE-interactions for membrane fusion give

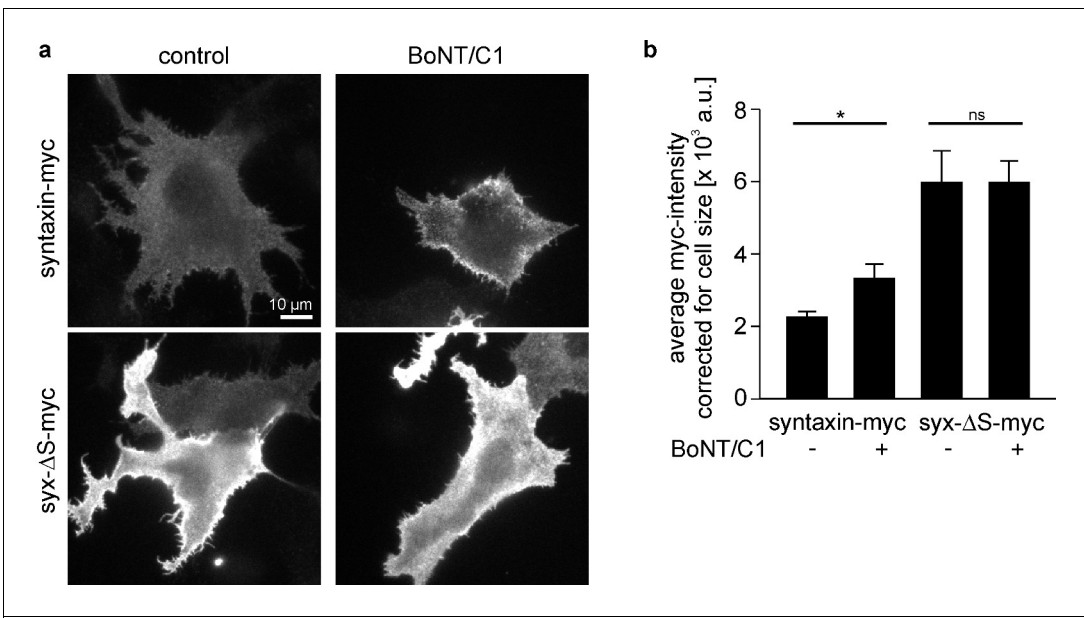

**Figure 7.** Myc-tag accessibility increases after intracellular cleavage of syntaxin. (**a**) Cells expressing syntaxin-myc or syx-ΔS-myc were co-transfected (BoNT/C1) or not (control) with the light chain of botulinum neurotoxin C1 (BoNT/C1) fused to GFP. One day after transfection cells were fixed, stained for myc and imaged for quantification of the myc- (**a**) and GFP-intensity (not shown). Images are shown at the same scalings. (**b**) Staining quantified from outlined cells. While BoNT/C1 did not change the size of syx-ΔS cells, it produced ≈ 30% smaller syntaxin-myc cells, indicating that syntaxin-myc cells may concentrate their fluorescence on smaller areas. This may lead to an overestimation of the BoNT/C1 induced increase in brightness. Therefore the mean intensities were related to the average size of the cells. Values are given as means ± S.E.M. (n = 5). t-test on syntaxin-myc/syntaxin-myc + BoNT/C1, p=0.034 (*p<0.05); t-test on syx-ΔS-myc/syx-ΔS-myc + BoNT/C1, p=0.998 (ns, not significant p>0.05).

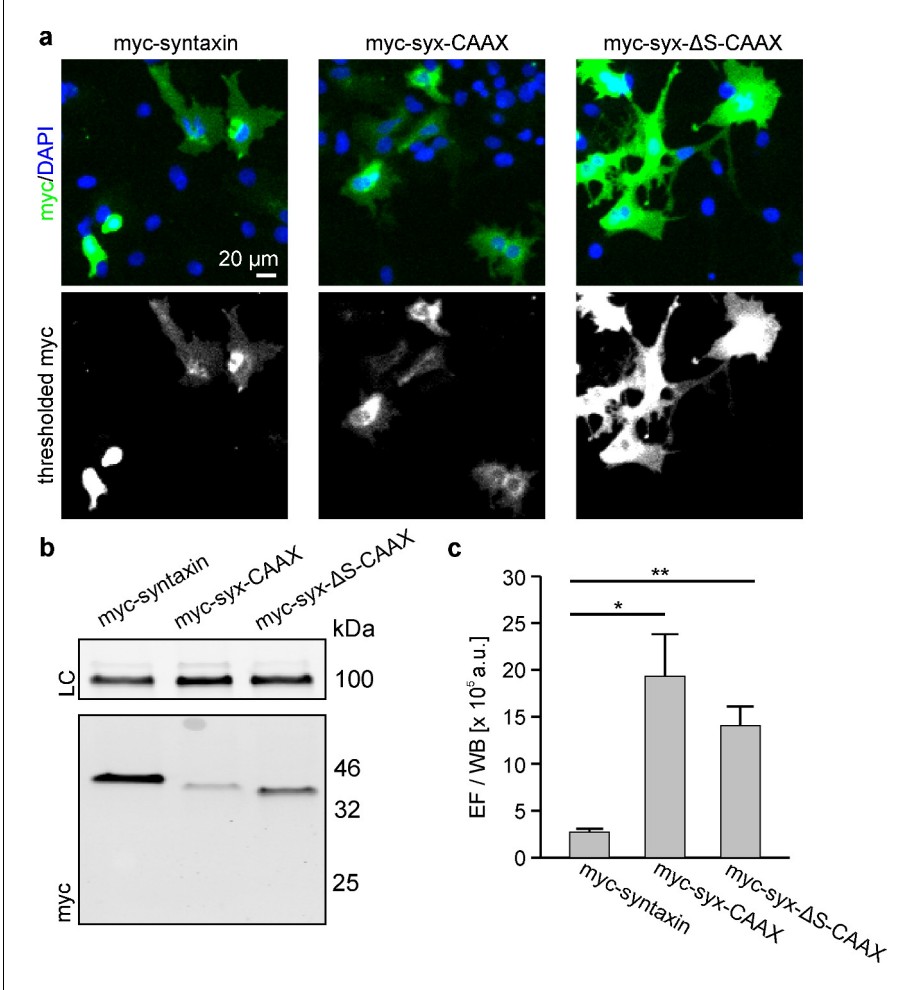

**Figure 8.** Exchange of the TMS by a CAAX-box decreases the packing density. Cells were transfected with plasmids encoding for N-terminally myc-tagged syntaxin (myc-syntaxin), myc-syx-CAAX or myc-syx-ΔS-CAAX (see *Figure 1* for details). Visualization on membrane sheets shows that also the CAAX-constructs form clusters (*Figure 8—figure supplement 1*). Constructs were analysed as described in *Figure 6*. (a) Microscopy images from fixed cells expressing the indicated constructs. Shown are magnified views from the images as overlays (blue, DAPI; green, myc-staining). (b) From the same transfections the western blot analysis is shown. Top, loading control (LC) using transferrin receptor staining; bottom, myc-signal. (c) Absolute epifluorescence signals (EF) related to the corresponding absolute western blot signals (WB). Values are given as means ± S.E.M. (n = 3), (t-test: p=0.021 for myc-syntaxin/myc-syx-CAAX; p=0.005 for myc-syntaxin/myc-syx-ΔS-CAAX; *p<0.05, **p<0.01).

The following figure supplement is available for figure 8:

**Figure supplement 1.** Visualization of myc-syntaxin, myc-syx-CAAX and myc-syx-ΔS-CAAX clusters on membrane sheets imaged by STED microscopy.

some hints. First, the SNARE-motif N-terminal portion, which we found to be crucial for mobility restriction in this study, is also required for zippering from the N- to the C-termini of a heterologous SNARE-motif bundle (*Sørensen et al., 2006*). A very similar mobility pattern as in this study was previously observed in neuronal PC12 cells that express high levels of endogenous syntaxin 1A along with its primary binding partner SNAP25. Here, syntaxin lacking the entire N-terminus and the linker region becomes proceedingly faster upon removal of more and more amino acids from the SNARE-motif (*Sieber et al., 2007*). Second, in the ternary SNARE complex syntaxin helicity extends into the membrane (*Stein et al., 2009*). Such helicity extension also seems to be important for SNARE-

clustering as indicated by our CAAX experiment. Although we cannot exclude that other, yet unidentified proteins participate in clustering, we propose the simplest core clustering reaction would be a homophilic or heterophilic SNARE-zipper. Because the syntaxin molecule carries a net negative charge, electrostatic repulsions likely preclude direct contact of all SNARE motifs. It is tempting to speculate whether such dense packing can be achieved in the presence of $Ca^{2+}$ that increases the packing density of negatively charged proteins, among them syntaxin 1A (*Zilly et al., 2011*).

In conclusion, two basic mechanisms underlie membrane protein clustering: loose clustering for which the TMS is sufficient, and cytoplasmic interactions which mediate tight cluster packing. Initially, loose-cluster formation is crucial, as without proper membrane propagated forces (e.g. cholesterol depletion) clusters disperse. In other words the cytoplasmic reaction alone is not sufficient for cluster formation, at least not for tightly packed clusters as suggested by the CAAX experiment. Because essentially all membrane domains depend on cholesterol (*Saka et al., 2014*), we propose that the here identified hierarchical model of membrane protein clustering is of general significance.

Moreover, the data show that size does not necessarily scale with the number of molecules the structure contains. It is not sufficient to know the representative distribution of components for the understanding of self-organization processes. Instead, a relation between size and the relative number of underlying molecules is required. Doing so we anticipate many self-organization processes could be further dissected, allowing an understanding of the architecture of cellular clusters and other supramolecular assemblies.

## Materials and methods

### Cloning

Syntaxin constructs are based on the sequence of rat syntaxin1A (NP_446240.2; SNARE motif and pbs are identical to human syntaxin 1A (NP_004594.1) differing in T21A, I278V, I283V and G288A). Syntaxin-GFP and syx-ΔCyt-GFP are described in *Sieber et al. (2006)* (corresponding to Sx1A-GFP and Sx1A, TMR-GFP, respectively). Syx-ΔS-GFP was obtained by fusion PCR using Syntaxin-GFP as a template. The restriction sites NheI and NotI were used for insertion of the PCR product into the backbone of the expression vector pEGFP-N1 (Clontech, Mountain View, GenBank accession number: U55761), where the EGFP carries a mutation to abolish GFP dimerization (*Zacharias et al., 2002*), in the following called pmEGFP-N1. For the C-terminal myc-tagged syntaxin variants, syntaxin-GFP and syx-ΔCyt-GFP with the C-terminal linker *LVPRARDPPVAT* were used as templates for PCR amplification generating HindIII and XbaI restriction sites. For syx-ΔS an additional fusion PCR step was performed before further subcloning of all three constructs into the vector pDL208 based on a pcDNA 3.1Hygro(+) where a triple myc tag was inserted (*Lohmann, 2013*). VAMP8-GFP was obtained by subcloning of VAMP8 from VAMP8-mCherry (kindly provided by Jens Rettig and described in *Marshall et al., 2015*) into the backbone of the pmEGFP-N1 vector using NheI and KpnI restriction sites. In GFP-SNAP25 a mEGFP is fused N-terminally to rat SNAP-25B (NP_112253.1) (*Halemani et al., 2010*). The BoNT/C1/LC-GFP (LC, light chain) plasmid is based on pmEGFP-N1 were the light chain of botulinum neurotoxin C1 (BoNT/C1; *Zilly et al., 2006*) was inserted using the EcoRI and KpnI restriction sites. The CAAX box was fused C-terminally to amino acid Q258. The constructs are based on an N-terminally myc-tagged syntaxin 1A (*Sieber et al., 2006*) used as a template for single step PCR in the case of myc-syx-CAAX and a fusion PCR for myc-syx-ΔS-CAAX. Using the NheI and NotI restriction sites, the PCR products were inserted into a vector containing the polybasic amino acid stretch and the CAAX box of K-Ras (AAAKSKTK*CVIM*; CAAX box in italic) fused to a RFP tagged synaptojanin (kindly provided by Silvio Rizzoli and described in *Malecz et al., 2000*; cut out beforehand by NheI and NotI restriction sites). All constructs were verified by sequencing.

### Cell culture and generation of membrane sheets

HepG2 cells (Cell Lines Service; CLS Cat# 300198; RRID: CVCL_0027) were maintained and propagated essentially as described previously (*Schreiber et al., 2012*). Cells were tested negative for mycoplasma (GATC Biotech, job number 95361). Cells were plated on glass coverslips coated with poly-L-lysine for fluorescence microscopy. Neon electroporation system (Thermo Fisher Scientific)

was used to transfect cells as described previously (*Schreiber et al., 2012*). Cells were used for experiments 1 day after transfection.

For membrane sheet generation, cells were subjected to a short ultrasound pulse in ice-cold sonication buffer (120 mM potassium glutamate, 20 mM potassium acetate, 20 mM HEPES-KOH, 10 mM EGTA; pH 7.2). For FRAP experiments membrane sheets were kept for imaging in sonication buffer. For STED microscopy membrane sheets were rinsed with PBS (136 mM NaCl, 2.7 mM KCl, 1.47 mM $KH_2PO_4$, 8.1 mM $Na_2HPO_4$), fixed and immunostained (see below).

## Epifluorescence microscopy

For epifluorescence microscopy an Olympus IX81-ZDC fluorescence microscope (*Zilly et al., 2011*) with a 60 × 1.49 NA apochromat objective or a 10 × 0.4 NA universal plan apochromat objective was used. For fluorescence excitation and detection we employed a MT20E Illumination system (Olympus) with a 150 W Xenon lamp and the filter sets F36-500 DAPI HC (for DAPI or TMA-DPH), F36-525 EGFP HC (for GFP or Alexa488), and F46-009 Cy5 ET (for Atto647N; all filters from AHF Analysentechnik, Tuebingen, Germany). For imaging, an EMCCD camera (ImagEM C9100-13, Hamamatsu Photonics, Hamamatsu, Japan) was used. Fixed samples were imaged in PBS containing TMA-DPH (ThermoFisherScientific, #T204) that visualizes membranes, or in PBS without TMA-DPH when DAPI (Sigma-Aldrich, #D9542) was used for staining of cell nuclei. Live cells in the GFP pH quenching experiments were imaged in phosphate salt solutions (different from PBS, see below). Images were analysed with ImageJ (for details see below) and are shown at arbitrary scaling, if not stated otherwise.

## pH-quenching of GFP

One day after transfection with plasmids encoding for either syntaxin-GFP, syx-ΔS-GFP, syx-ΔCyt-GFP, GFP-SNAP25 or VAMP8-GFP cells were treated with 0.2 µM bafilomycin A1 (diluted from a stock solution in DMSO, yielding 1% DMSO in the working solution) from *Streptomyces griseus* (Sigma-Aldrich, #B1793) or in controls with DMSO alone (1%) for 45 min at 37°C. Using the 10x objective several cells were imaged in phosphate buffer pH7.4 (30.5 mM sodium chloride, 81 mM $Na_2HPO_4$, 19 mM $NaH_2PO_4$). Buffer was exchanged to phosphate salt solution pH4.3 (30.5 mM sodium chloride, 8 mM $Na_2HPO_4$, 92 mM $NaH_2PO_4$) by three brief washes with the new solution, followed by a 2–3 min incubation. Then the same cells were imaged again.

For analysis, in ImageJ cells were outlined manually and mean fluorescence intensity in the generated ROIs was corrected for local background. For every cell, the intensity value at pH 4.3 was related to the intensity at pH 7.4.

## FRAP

For fluorescence recovery after photobleaching (FRAP) experiments HepG2 cells were transfected with plasmids encoding for syntaxin-GFP, syx-ΔS-GFP or syx-ΔCyt-GFP. Experiments were performed at 37°C. Cells were analysed in Ringer solution (130 mM NaCl, 4 mM KCl, 1 mM $CaCl_2$, 1 mM $MgCl_2$, 48 mM D(+)αGlucose, 10 mM HEPES pH7.4) and membrane sheets in sonication buffer. For imaging we used an Olympus Fluoview 1000 laser scanning microscope (*Zilly et al., 2011*) adjusting the pixel size to 414 nm and ROIs for bleaching were 12 pixel x 12 pixel. In order to have samples with comparable intensities, bright cells or membrane sheets with high levels of syx-ΔS-GFP and syx-ΔCyt-GFP were not selected for imaging. Bleaching was performed using the 488 nm and 405 nm laser at full intensities. Recordings were taken at 1 Hz, starting with a 10 picture pre-bleach sequence, followed by a bleach event for 500 ms and a 200 s post-bleach sequence. Per mounted coverslip up to 4 cells or membrane sheets were analysed. For membrane sheets we analysed for syntaxin-GFP whether there is a correlation between the sequence number (first to last recorded membrane sheet, which correlates with the time after membrane sheet generation) and the half-time of recovery, which was not the case.

Analysis of the experiments was done essentially as published in *Zilly et al. (2011)*. In brief, to detect out of focus drift a control region in a non-bleached area was used comparing background corrected fluorescence of the first 10 frames to the last 3 frames. If intensity differed by more than ±15% the measurement was excluded from further analysis. Intensity recovery traces obtained from the ROIs were background corrected and all traces from one experiment were averaged. The average trace was fitted to a hyperbolic function with an offset obtaining the half time of recovery.

For plotting intensity versus half time of recovery, hyperbola functions were fitted to traces from individual cells. Fits with a lower $R^2$ than 0.7 were omitted from the analysis, still leaving 5–13 cells or 5–15 membrane sheets per construct and experiment.

## Immunostaining

Cells or membrane sheets were fixed in 4% PFA in PBS for 30–45 min at RT. PFA was quenched with 50 mM $NH_4Cl$ in PBS. Samples were blocked with 3% BSA in PBS for 1–2 hr at RT prior to immunostaining. Monoclonal mouse antibodies were used for the detection of the myc-tag (ThermoFisherScientific, Cat# MA1-21316, RRID: AB_558473) and syntaxin (HPC1; Sigma, #S0664, RRID: AB_477483; used only in pulsed STED experiments). GFP was detected employing a polyclonal rabbit antibody (Abcam, #ab290, RRID: AB_303395). Incubation with the first antibody diluted 1:200 or 1:400 in 3% BSA in PBS lasted 1.5 hr at RT for the BoNT/C1 experiment (as secondary antibody Atto647N labeled goat-anti-mouse (Sigma, #50185) was used; all secondary antibodies were diluted 1:200), overnight at 4°C for experiments in which myc-tag staining was related to western blot (secondary antibodies were Alexa488 labelled goat-anti-mouse (ThermoFisherScientific, #A11001) or donkey-anti-mouse (ThermoFisherScientific, #A21202)) or pulsed STED microscopy on membrane sheets (as secondary antibody Atto647N labeled goat-anti-mouse (Sigma, #50185) was used), for 24 hr at 4°C for CW-STED microscopy on membrane sheets (secondary antibody was Alexa488 labelled goat anti-mouse (ThermoFisherScientific, #A11001)) or 2 hr for STED microscopy on cells (secondary antibody was Atto647N labelled goat-anti-mouse (Sigma, #50185) or Atto647N labelled goat-anti-rabbit (Sigma, #40839)), followed by 3 washing steps with PBS. Secondary antibody incubation was performed at RT for 1 hr for the BoNT/C1 experiments, the experiments in which the myc-tag staining was related to WB signals and STED experiments on cells, 2 hr for pulsed STED microscopy on membrane sheets and overnight at 4°C for CW-STED microscopy on membrane sheets likewise in 3% BSA in PBS again followed by 3 washing steps with PBS. For STED microscopy on immunostained membrane sheets samples were embedded with ProLong Gold Antifade Mountant (ThermoFisher Scientific, #P10144) using microscopy slides, sealed and stored at 4°C prior to imaging.

## STED microscopy

A TCS-SP8 gated-STED microscope (Leica, Mannheim, Germany; available at the DZNE imaging facility, located at the CAESAR, Bonn, Germany) was used for CW-STED high resolution imaging of myc-stained membrane sheets. Images were recorded at 200 Hz exciting at 488 nm (using a white-light laser at 20% power), depleting with a 592 nm CW-STED beam (at 50% intensity), detecting at 495-570 nm with time-gating from 1 to 6.5 ns (using a hybrid detector), and with a pixel size of 20 nm.

Pulsed STED high resolution imaging on membrane sheets and fixed cells was performed on myc- or HPC1-stained samples using a four-channel confocal laser scanning microscope equipped with an easy-3D STED module (Abberior Instruments; Göttingen, Germany; available at the LIMES institute imaging facility, Bonn, Germany). Fast DiO membrane stain images were recorded exciting with a pulsed 488 nm laser (0.2 mW) and detecting from 500 to 520 nm and 532–558 nm via avalanche photo diodes (APD). For Super-resolution images, Atto-647 was excited using a pulsed 640 nm laser (0.3 mW), depleted using a pulsed 775 nm laser (350 or 450 mW) and detected at 650–720 nm with time-gating from 1.25 to 9.25 ns (using an APD), and with a pixel size of 20 × 20 nm.

Spot analysis was performed employing a custom ImageJ macro. Using the ImageJ function 'Find Maxima', Syntaxin spots were identified in DOG (difference of Gaussians)-filtered STED images within squared ROIs (180 pixels x 180 pixels and 130 pixel x 130 pixel for CW-STED and pulsed-STED images, respectively). Further analysis of the spot features was performed on raw STED images placing circular ROIs (5 pixel diameter) at the identified spot positions. Weak spots with a low mean intensity (100 or 4 a.u. for CW and pulsed STED images respectively) were excluded. Extracted parameters were the mean intensity of the spot ROI, the spot size based on a Gaussian fit of a line scan (15 × 3 pixels, horizontal or vertical depending on fit quality) and the nearest neighbour distance based on the centre of mass within spot ROIs. Subsequently, spot parameters were averaged for each membrane sheet and related to the background-corrected mean of the myc- or HPC1-intensity of the membrane sheets in the confocal channel (recorded simultaneously with the STED image). For calculating mean spot intensities, all spots were included, for the mean size only spots with at least 5–6 pixels nearest neighbour distance and suitable Gaussian fit quality ($R^2 \geq 0.9$, assuring a well-centred peak, reasonable offset and

peak height). For CW-STED, plots show mean sheet values for three independent experiments with a total of 38–39 sheets (2924–3185 spots) per condition for cluster size and 39–49 sheets (4678–9385 spots) for cluster counts and intensities. For pulsed STED, three independent experiments were included with a total of 22–45 sheets (1336–2010 spots) per condition for cluster size and 33–48 sheets (2407–6054 spots) for cluster counts and intensities.

Only for STED microscopy on whole cells, fixed cells were labelled with 100 µg/mL concanavalin A conjugated to Alexa594 (ThermoFisher Scientific, # C11253) in HBSS (Hanks' Balanced Salt Solution; ThermoFisher Scientific, #14175053) for 30 min followed by permeabilisation for immunostaining with 0.5% Triton-X100 in PBS for 10 min prior to the blocking step. During immunostaining all solutions contained 0.05% Triton-X100. Imaging was performed in PBS on the easy3D STED microscope using a water immersion objective. Samples were screened for transfected cells in the epifluorescence mode of the microscope. Cells were then recorded in confocal mode in the x,y-plane for an overview image (110 × 110 µm, pixel size 300 nm) of the concanavalin and the antibody staining. In the overview image horizontally 10 x,z-sections were defined for scanning, with a distance of 10 µm, which were imaged in the concanavalin and in the antibody channels employing the 3D-STED modus of the microscope (starting the scan at the basal membrane; for illustration images were rotated by 180°) and 200 mW STED laser deexcitation power (100% power distribution to the 3D-STED donut) at a pixel size of 50 nm in x and 96.3 nm in z. For the concanavalin channel, Alexa-594 was excited using a pulsed 561 nm laser (0.03 mW) and detected at 605-625 nm with time-gating from 0.78 to 8.78 ns via an APD. For the antibody channel, Atto647N was excited using a pulsed 640 nm laser (0.1 mW) and detected at 650-720 nm with the same time gating. For STED images, 6% Alexa-594 bleed-through into the Atto647 image was found in single stains (concanavalin A-Alexa594 only), which was subtracted from the Atto647 recordings.

For analysis, the cell membrane was traced by a ROI in the concanavalin A image in each slice of one cell. This ROI was expanded by five pixels defining an inner ROI delineating the cytosol and an outer ROI delineating the whole cell. ROIs were then transferred to the corresponding STED images of the antibody staining. The integral of the fluorescence signal was measured in the cytosol and the whole cell ROIs and was summed up for the slices of one cell. Averaged values of non transfected control cells were subtracted for background correction and the value for the cytosolic signal was related to the total fluorescence signal of the respective cell. Images are shown at arbitrary scaling, if not stated otherwise.

## Comparison epifluorescence and western blot

For the comparison of microscopic fluorescence and western blot signals HepG2 cells were transfected with plasmids encoding for syntaxin-myc or syx-ΔS-myc using different plasmid concentrations (a low concentration of 10 µg per 4 × 10⁶ cells for both constructs; a high concentration of 20 µg or 30 µg per 4 × 10⁶ cells for syntaxin-myc and syx-ΔS-myc, respectively). This yielded a 'low' and 'high' pair of syntaxin-myc/syx-ΔS-myc transfected cells (the plasmid concentrations were adjusted to get for each pair a roughly similar expression level in the western blot). In separate experiments cells were transfected with plasmids encoding for myc-syntaxin, myc-syx-CAAX or myc-ΔS-CAAX using for each construct 10 µg plasmid DNA per 4 × 10⁶ cells.

After the transfection, cells were split into two fractions: one for coverslips for microscopy and the other part was transferred to a cell culture dish for western blot analysis. After one day, cells were immunostained as described above. Prior to the blocking step they were permeabilized with 0.2% Triton X-100 in PBS for 10 min, and 0.2% Triton X-100 was added to all solutions during immunostaining. Prior to imaging, cells were stained with DAPI (0.1 µg/ml) (Sigma-Aldrich, # D9542) in PBS for 10 min at RT followed by three washing steps with PBS. For imaging, cells were selected in the DAPI channel followed by taking images in the DAPI channel and the green channel (recording Alexa488). For analysis, a threshold of background signal in the green channel was manually determined from mock transfected cells that underwent the same staining protocol. The threshold value was subtracted from all images prior to the calculation of their integrated fluorescence intensity. For each image, the integrated intensities were divided by the number of cells in the corresponding images which was obtained by counting the DAPI stained nuclei by the 'Find Maxima' tool in ImageJ. Bright DAPI aggregates were distinguished from nuclei by an intensity threshold. For each transfection and construct, the values from all images were averaged yielding the value 'EF' (epifluorescence).

In parallel, for western blot analysis cells were scraped off the flask in ice-cold PBS, centrifuged and lysed in RIPA buffer (Santa Cruz, #sc-24948) for 30 min on ice, intermitted by two 5 min incubations in an ice cold ultrasound bath. Lysates were centrifuged at 14,000 x g for 15 min. Protein content of the supernatant was determined using a BCA assay (Thermo Fisher Scientific, #23225). 4x Laemmli buffer was added to 5 to 25 µg of total protein per lane and samples were heated for 10 min at 95°C. Samples in Laemmli buffer were subjected to SDS-PAGE. A wet electroblotting system (BioRad) was used to transfer proteins to a nitrocellulose membrane (Carl Roth, Germany). For the detection of the myc-tag a mouse monoclonal antibody was used (ThermoFisherScientific, Cat# MA1-21316, RRID: AB_558473), transferrin receptor was detected with a rabbit polyclonal antibody (abcam, Cat# ab84036, RRID: AB_10673794). Prior to the blocking step and antibody incubation, the membrane was cut into an upper and lower part for separate detection of transferrin receptor and myc-tag, respectively. Blocking was performed at RT in Odyssey Blocking Buffer (P/N 927–4000, Licor Bioscience) diluted with PBS in a 1:1 ratio. First antibody (1:1000 dilution) incubation in Odyssey Blocking Buffer with PBS/0.1%Tween20 (PBS-T) (1:1) was over night at 4°C. Membranes were washed 4 times for 5 min with PBS-T prior to the incubation with the secondary antibodies (using goat-anti-mouse-IRDye800CW (Odyssey, #926–32210; diluted 1:10,000) and goat-anti-rabbit-IRDye800CW (Odyssey, #926–32211; diluted 1:15,000) diluted in Odyssey Blocking Buffer with PBS-T (1:1) for 1 hr at RT followed by four washing steps in PBS-T for 5 min. Membranes were washed with PBS to remove the detergent prior to imaging with an Odyssey near-infrared imaging system (LI-COR Biosciences). Analysis of western blots was performed using the gel analyser plugin in ImageJ. Peak areas were related to the corresponding signal of the transferrin receptor loading control from the same lane, yielding for each transfection and construct the value 'WB'. For the calculation of the signal ratios, EF was divided by WB.

## Botulinum toxin co-expression

HepG2 cells were transfected with 10 µg of plasmid encoding for syntaxin-myc or syx-ΔS-myc only or together with a plasmid encoding the light chain of botulinum neurotoxin C1 C-terminally fused to GFP (BoNT/C1-GFP). To keep BoNT/C1-GFP expression low, its plasmid was used at a 10-fold lower concentration (1 µg per transfection). After 1 day cells were fixed, stained for the myc-tag (anti-myc/goat-anti-mouse-Atto647N) and imaged with an epifluorescence microscope in the presence of TMA-DPH in the blue (TMA-DPH), green (GFP) and the far-red (Atto647N) channel. For analysis, cell footprints were outlined in the myc-channel creating a ROI for measuring size and fluorescence intensity in the GFP as well as the myc-channel. Intensities were corrected for background fluorescence. From the single transfected cells (no BoNT/C1-GFP) the mean autofluorescence in the GFP channel was calculated. Cells from the double transfection with lower values in the GFP channel than the calculated autofluorescence were considered not to contain BoNT/C1-GFP and thus were omitted from further analysis. Since cell size changed among the conditions (especially upon co-transfection of syntaxin-myc and BoNT/C1-LC-GFP cell size decreased), the intensity of the myc-signal was corrected for variations in the average size of the cells (22–53 cells per condition and experiment).

## Acknowledgements

The authors would like to express their gratitude to Jens Rettig (Homburg, Germany), Daniel Lohmann (Bonn, Germany) and Silvio O. Rizzoli (Göttingen, Germany) for providing plasmids. We thank the DZNE in Bonn for providing access to a CW STED microscope in their imaging facility. This work was supported by a grant from the Deutsche Forschungsgemeinschaft (TRR83).

## Additional information

### Funding

| Funder | Grant reference number | Author |
| --- | --- | --- |
| Deutsche Forschungsgemeinschaft | TRR83 | Thorsten Lang |

The funders had no role in study design, data collection and interpretation, or the decision to submit the work for publication.

## Author contributions

EM, Conceptualization, Formal analysis, Validation, Investigation, Visualization, Methodology, Writing—original draft, Writing—review and editing; J-GS, Conceptualization, Formal analysis, Validation, Investigation, Visualization, Methodology, Writing—original draft; PW, HB, Formal analysis, Validation, Investigation, Writing—review and editing; SH, Validation, Investigation, Methodology; NK, Investigation, Writing—original draft, Writing—review and editing; JF, Formal analysis, Writing—review and editing; TL, Conceptualization, Supervision, Funding acquisition, Writing—original draft, Project administration, Writing—review and editing

## Author ORCIDs

Elisa Merklinger, http://orcid.org/0000-0002-7257-1895
Pascal Weber, http://orcid.org/0000-0002-6369-2708
Thorsten Lang, http://orcid.org/0000-0002-9128-0137

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
