## [Decision Letter]

Thank you for submitting your article "The packing density of a supramolecular membrane protein cluster is controlled by cytoplasmic interactions" for consideration by *eLife*. Your article has been favorably evaluated by Richard Aldrich (Senior Editor) and three reviewers, one of whom is a member of our Board of Reviewing Editors. The following individual involved in review of your submission has agreed to reveal their identity: Rory Duncan (Reviewer #2).

The reviewers have discussed the reviews with one another. Generally, all reviewers agree that your study is interesting and, in principle merits publication in *eLife*. However, some revisions are required, and the Reviewing Editor has drafted this decision to help you prepare a revised submission.

Summary:

This manuscript addresses the packing density of proteins within protein clusters in the plasma membrane. As an example the authors use the well-studied SNARE syntaxin1 for which clustering was shown previously to be mediated by a combination of independent interactions involving the transmembrane domain, positively charged clusters, and interactions between the cytoplasmic parts of the proteins. Using syntaxin mutants containing C-terminal myc-tags, the authors show that full-length syntaxin 1 is less accessible to antibody binding compared to deletion mutants due to steric hindrance within the domains. These data prove that the SNARE domain contributes to the clustering of syx1 in membrane domains, and the authors propose a model where both membrane and cytosolic protein interactions contribute to syx1 clustering. Moreover, using super-resolution microscopy, the authors confirm that the average number of molecules within a given cluster appears, at least to a certain extent, to be independent of the expression level, with higher expression levels yielding more instead of larger clusters.

Essential revisions:

1) Much of the data (Figure 4–Figure 6) relies on the assumption that 100% of the GFP- and myc-tagged syx1 mutants locate to the plasma membrane. If this is not the case and a fraction of the syx1 mutants are retained at ER/Golgi or sorted to endo/lysosomes, the assumption that the decreased GFP fluorescence is due to self-quenching does not hold. The same is true for the experiments with the myc-tagged proteins, where the reduced antibody binding can be equally well explained by intracellular pools of protein. This is especially a concern since Munc18-1, which is probably not expressed by HepG2 cells, might be required for targeting of syx1 to the plasma membrane (Rowe et al., JCS, 1999, 112: 1865). For the GFP-tagged constructs, the plasma membrane localization is shown in Figure 2—figure supplement 1. However, these experiments are not conclusive, as syx1-GFP in acidic organelles will already be quenched and this population cannot be accounted for. In addition, the cells will die at pH4.3 resulting in dissipation of membrane gradients and quenching of intracellular pools. To make these experiments more convincing, the authors should include control experiments with bafilomycin/NH4Cl, and show 3D confocal stacks with antibody labeling of syx1 and a plasma membrane protein. Control experiments with a transmembrane SNARE residing in intracellular compartments should also be included. To prove that the decreased fluorescence is really due to self-quenching, photobleaching curves where the fluorescence should initially increase should be provided. Finally, the 100% plasma membrane localization should be demonstrated for the myc-tagged syx1 mutants.

2) Does the recovery after photobleaching proceed at near identical rates in intact cell membranes versus sheets? These data are easy to acquire and analyse and would give some support to the ideas in the paper. Further, 1 Hz is very slow for FRAP recover curves and slow rates could mask multi-component behaviors. If possible I would like to see a comparison as a control with significantly higher sampling rates.

---

## [Author Response]

*Essential revisions:*

*1) Much of the data (Figure 4–Figure 6) relies on the assumption that 100% of the GFP- and myc-tagged syx1 mutants locate to the plasma membrane. If this is not the case and a fraction of the syx1 mutants are retained at ER/Golgi or sorted to endo/lysosomes, the assumption that the decreased GFP fluorescence is due to self-quenching does not hold. The same is true for the experiments with the myc-tagged proteins, where the reduced antibody binding can be equally well explained by intracellular pools of protein. This is especially a concern since Munc18-1, which is probably not expressed by HepG2 cells, might be required for targeting of syx1 to the plasma membrane (Rowe et al., JCS, 1999, 112: 1865). For the GFP-tagged constructs, the plasma membrane localization is shown in Figure 2—figure supplement 1. However, these experiments are not conclusive, as syx1-GFP in acidic organelles will already be quenched and this population cannot be accounted for. In addition, the cells will die at pH4.3 resulting in dissipation of membrane gradients and quenching of intracellular pools. To make these experiments more convincing, the authors should include control experiments with bafilomycin/NH4Cl, and show 3D confocal stacks with antibody labeling of syx1 and a plasma membrane protein. Control experiments with a transmembrane SNARE residing in intracellular compartments should also be included.*

We agree that for our conclusions we need to rule out large differences in plasma membrane targeting. Please note that we did not assume 100% targeting efficiency. In any case, we very much agree that the experiment shown in old Figure 2—figure supplement 1, suggesting plasma membrane targeting from 75 – 85% for the constructs, is not conclusive for two reasons. First, as already pointed out by the referees we could have intracellular pH-quenched fractions. Second, we probed quenching distal from the nucleus at locations we also would have bleached regions for FRAP, so we per se underestimate intracellular fractions.

As suggested, using VAMP8 as a reference for an intracellular SNARE we measured pH-induced GFP-quenching after bafilomycin treatment (new Figure 2 replacing old Figure 2—figure supplement 1). Bafilomycin was applied for 45 min at 37°C, a condition we found out sufficient to reach a plateau in VAMP8-GFP dequenching while still preserving cell integrity (although different from our old experiment we noticed a small drop of 15% for the control protein GFP-SNAP25, perhaps due to the addition of the Bafilomycin solvent DMSO). As suggested in the referee’s roadmap, we also tried ammonium chloride. However, this treatment tended to make cells permeable to acidic solution causing at pH4.3 a 50%-drop in fluorescence for the control protein GFP-SNAP25.

We further analysed the subcellular distribution of both GFP- and myc-tagged constructs by optical sectioning microscopy. For this approach, it was necessary to apply super-resolution STED microscopy to separate the ‘sandwiched’ plasma membranes at the cell periphery (see also Figure 9).

Author response image 1.Subcellular distribution analysis by 3D-STED microscopy.Cells expressing either syntaxin-GFP, syx-ΔS-GFP, syx-ΔCyt-GFP, VAMP8-GFP, syntaxin-myc, syx-ΔS-myc or syx-ΔCyt-myc were fixed and permeabilized prior to immunostaining with an antibody raised against the respective tag. The plasma membrane was counterstained with concanavalin A (not shown). (**a**) Images from the GFP-labelled constructs. The lower left insets show an x-y overview. Using this overview as reference an x-z slice was imaged every 10 µm by two colour 3D STED microscopy (using Alexa594 for concanavalin and Atto647N for the antibody labeling). (**b**) For analysis, in the concanavalin channel ROIs differentiating between the whole cell and the cytosol were generated. The integral of fluorescence was summed up for all slices from one cell. Background values from non-transfected cells were subtracted and the intracellular integrated signal from the cytosol was related to the whole cell signal for each cell. Values are given as means ± SEM (n = 2 – 3 independent experiments with 7 – 26 cells per condition).**DOI:**
http://dx.doi.org/10.7554/eLife.20705.014

Similar to old Figure 2—figure supplement 1, both methods do not show dramatically different sorting efficiencies to the plasma membrane. Compared to our old estimate, the new values for the GFP-tagged constructs indicated a 10 – 15% higher intracellular pool; most likely as in the old experiment the intracellular pools were neglected.

However, both approaches yield no consistent picture with respect to the exact size of the subcellular fractions. This becomes particularly clear when viewing the data for VAMP8. While the pH-quenching experiment show reasonable values for VAMP8 that are clearly different from the syntaxin constructs, optical sectioning microscopy seems incapable in resolving any clear differences. This might be primarily a problem of the analysis, since VAMP8 images suggest a large pool of spotty and bright intracellular fluorescence (see lower right panel of Figure 9), which is only minimally reflected in the analysis.

As exact values for calibration of membrane sheet fluorescence are essential, we communicated the problem to the editors, who indicated they were open for an alternative method. We turned to an approach that would not rely on the selective analysis of the plasma membrane fraction but instead probes the accessibility of the myc-tag in the whole cell. To avoid time consuming 3D analysis (which is difficult to correct for bleaching artefacts and apparently not very sensitive in detecting differences in plasma membrane targeting) we used a low magnification 10x objective with a depth of field of 8.5 μm capturing the fluorescence of the entire cell in the focal plane (for images see new Figure 6). After setting a threshold to eliminate signals from untransfected cells, the staining intensity of all transfected cells is measured and related to the number of cells (untransfected and transfected ones) present in this condition (the method allowed for a large sample number from 1,000 – 2,000 cells per condition in one independent experiment). This yields the average myc-stain in fluorescence microscopy per cell. From the same preparation a western blot analysis is performed as well. On the western blot membrane, the myc-tag is detected and related to the protein concentration using transferrin receptor as a loading control. As in the microscopy approach, the myc-signal is thereby related to all cells. Because the transfection rates are identical in both parts of the experiment, the microscopy and the western blot signals can be directly related to each other. Compared to syntaxin we find an approximately 3-fold brighter staining for the myc-epitope when attached to syx-ΔS. This value is in the same range as our old estimate using the IRES-system, which we have now removed from the manuscript because (i) of the uncertainty of the exact size of the plasma membrane pool, (ii) the criticism referring to too much data based on membrane sheets, and (iii) the criticism referring to the staining gradient. The data has been incorporated as new Figure 6 and is described in the seventh paragraph of the Results.

*To prove that the decreased fluorescence is really due to self-quenching, photobleaching curves where the fluorescence should initially increase should be provided.*

As shown in the new Figure 2, there could be a small fraction of intracellular pH sensitive syntaxin-GFP but not syx-ΔS-GFP, which may at least in part contribute to the GFP-quenching effect. Due to time restrictions, we did not repeat the FACS experiment in the presence of bafilomycin. We would, however, expect that the difference is diminished (or hardly detectable) leading to results which are no longer conclusive. Therefore, we have removed the FACS experiment from the manuscript.

*Finally, the 100% plasma membrane localization should be demonstrated for the myc-tagged syx1 mutants.*

Please see our reply above.

*2) Does the recovery after photobleaching proceed at near identical rates in intact cell membranes versus sheets? These data are easy to acquire and analyse and would give some support to the ideas in the paper.*

We now included FRAP experiments on membrane sheets (new Figure 4, Results, third paragraph). The findings are very similar to intact cells, supporting the idea of the paper.

*Further, 1 Hz is very slow for FRAP recover curves and slow rates could mask multi-component behaviors. If possible I would like to see a comparison as a control with significantly higher sampling rates*.

A freely diffusing single-span membrane protein has a diffusion coefficient of 0.23 μm^[3]^/s (Kenworthy et al., J. Cell Biol., 2004; please note this is just one example and values between publications can differ). With the ROI size of roughly 5 μm x 5 μm used in our experiments, for a protein with the above mentioned diffusion coefficient the half-time is about 7s (which is in the range of what we observe for the fastest construct). We therefore thought a sampling rate of 1Hz would be sufficient. However, to ensure we aren’t missing anything, and as requested by the referees, we repeated the FRAP measurements with a three-fold higher sampling rate. The data shows no difference in the recovery traces imaged at 1 Hz and 3 Hz (please see Figure 10).

Author response image 2.Recovery curves from syntaxin-GFP from sequences imaged at 1 Hz and 3 Hz.Same experiment as shown in Figure 3 (previous Figure 2). In brief, live cells expressing syntaxin-GFP were imaged by confocal microscopy at the basal plasma membrane. The fluorescence was bleached in a squared ROI and the recovery of fluorescence was monitored for several minutes either at 1 Hz (left panel, light brown) or at 3 Hz (middle panel, dark brown). Shown are averaged recovery traces from 10-11 cells. The traces were fitted with a hyperbolic function (solid line). The right panel shows an overlay of the two traces. Values are given as mean ± SD.**DOI:**
http://dx.doi.org/10.7554/eLife.20705.015